# Repositioning the Subject within Image

**Yikai Wang**[1] **Chenjie Cao**[1,2,3] **Ke Fan**[1] **Qiaole Dong**[1] **Yifan Li**[1] **Xiangyang Xue**[1] **Yanwei Fu**[1]
[1]*Fudan University;* [2]*DAMO Academy, Alibaba Group;* [3]*Hupan Lab*
*yi-kai.wang@outlook.com, yanweifu@fudan.edu.cn*

**Reviewed on OpenReview:** *https://openreview.net/forum?id=orHH4fCtR8*

## Abstract

Current image manipulation primarily centers on static manipulation, such as replacing specific regions within an image or altering its overall style. In this paper, we introduce an innovative dynamic manipulation task, subject repositioning. This task involves relocating a user-specified subject to a desired position while preserving the image's fidelity. Our research reveals that the fundamental sub-tasks of subject repositioning, which include filling the void left by the repositioned subject, reconstructing obscured portions of the subject and blending the subject to be consistent with surrounding areas, can be effectively reformulated as a unified, prompt-guided inpainting task. Consequently, we can employ a single diffusion generative model to address these sub-tasks using various task prompts learned through our proposed task inversion technique. Additionally, we integrate pre-processing and post-processing techniques to further enhance the quality of subject repositioning. These elements together form our SEgment-gEnerate-and-bLEnd (SEELE) framework. To assess SEELE's effectiveness in subject repositioning, we assemble a real-world subject repositioning dataset called ReS. Results of SEELE on ReS demonstrate its efficacy. Code and ReS dataset are available at `https://yikai-wang.github.io/seele/`.

## 1 Introduction

In 2023, Google Photos introduced an AI editing feature allowing users to reposition subjects within their images (Google, 2023). However, a lack of technical documentation limits understanding of this feature. Some researches have touched on aspects of it. Iizuka et al. (2014) explored object repositioning before the deep learning era, using user inputs like ground regions and bounding boxes. In the deep learning era, fields like scene decomposition (Zheng et al., 2021) and de-occlusion (Zhan et al., 2020) enable manipulation of object positions after the explicit understanding of scene and object relationships. This paper addresses general Subject Repositioning (SubRep) task without explicit scene understanding. Our aim is to address SubRep via a meticulously crafted solution, driven by a single diffusion model.

From an academic perspective, this task falls under image manipulation (Gatys et al., 2016; Isola et al., 2017; Zhu et al., 2017; Wang et al., 2018; El-Nouby et al., 2019; Fu et al., 2020; Zhang et al., 2021). Recent advancements in large-scale generative models have fueled interest in this field. These models, including generative adversarial models (Goodfellow et al., 2014), variational autoencoders (Kingma & Welling, 2014), auto-regressive models (Vaswani et al., 2017), and notably, diffusion models (Sohl-Dickstein et al., 2015), demonstrate impressive image manipulation capabilities with expanding model architectures and training datasets (Rombach et al., 2022; Kawar et al., 2022; Chang et al., 2023). However, current image manipulation methods primarily target "static" alterations, modifying specific image regions using cues like natural language, sketches, or layouts (El-Nouby et al., 2019; Zhang et al., 2021; Fu et al., 2020). Another aspect involves style-transfer tasks, transforming overall image styles such as converting photos into anime pictures or paintings (Chen et al., 2018; Wang et al., 2018; Jiang et al., 2021). Some extend to video manipulation, altering style or subjects over time (Kim et al., 2019; Xu et al., 2019; Fu et al., 2022). In contrast, subject repositioning dynamically relocates selected subjects within a single image while leaving the rest unchanged.

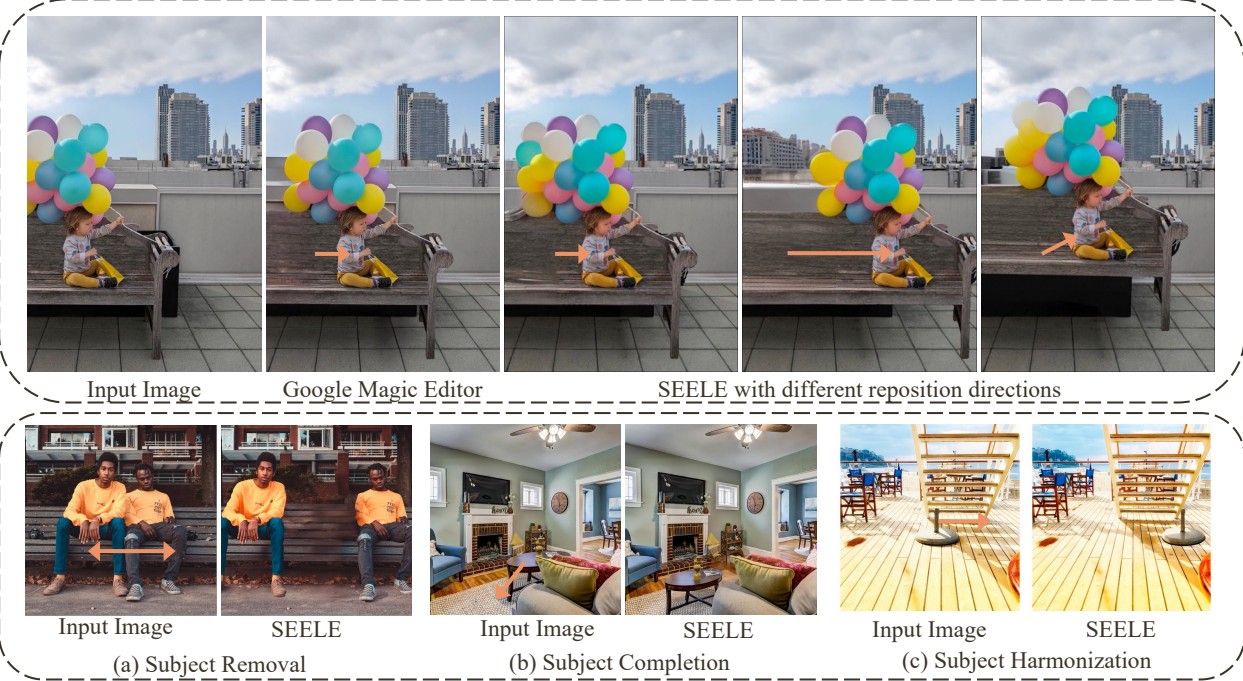

Figure 1: We compare subject repositioning using our SEELE with Google's Magic Editor. SEELE effectively addresses tasks like subject removal, completion, and harmonization through a unified prompt-guided inpainting process, powered by a single diffusion model. Comprehensive results are depicted in Figure 5.

The SubRep task involves multiple stages, including non-generative and generative tasks. Existing pretrained models are effective for non-generative tasks like segmenting subjects (Kirillov et al., 2023) and estimating occlusion relationships (Ranftl et al., 2020). Our focus lies on the generative tasks of SubRep, including: i) *Subject removal*: The generative model must fill voids left after repositioning without introducing new elements. ii) *Subject completion*: If the repositioned subject is partially obscured, the model must complete it to maintain integrity. iii) *Subject harmonization*: The repositioned subject should blend with surrounding areas. All these sub-tasks demand unique generative capabilities.

The most powerful text-to-image diffusion models (Nichol et al., 2022; Ho et al., 2022; Saharia et al., 2022; Ramesh et al., 2022; Rombach et al., 2022) show potential promise for SubRep. However, a key challenge is finding suitable text prompts, as these models are usually trained with image captions rather than task-specific instructions. The best prompts are often image-dependent and hard to generalize, limiting practical use in real-world applications. Translating these task instructions into caption-style prompts for fixed text-to-image diffusion models is particularly challenging. On the other hand, specialized models exist for specific aspects (Figure 1) of SubRep, like local inpainting (Zeng et al., 2020; Zhao et al., 2021; Li et al., 2022; Suvorov et al., 2022; Dong et al., 2022), subject completion (Zhan et al., 2020), and local harmonization (Xu et al., 2017; Zhang et al., 2020; Tsai et al., 2017). However, combining components from these models can make the SubRep system bulky and less elegant. Given the shared generative nature of these sub-tasks, our study raises an intriguing question: "Can we achieve all these sub-tasks using a single model?"

To answer this question, we introduce "task inversion", a novel concept that learns latent embeddings as alternative of text conditions to guide diffusion models with specific task instructions. The embedding space of text prompts in diffusion models offers versatility beyond just captions. Employing prompt tuning at the task level allows us to learn latent embeddings to guide diffusion models based on task instructions. Task inversion enables diffusion models to adapt to various tasks by adjusting task-level "text" prompts. Unlike textual inversion (Gal et al., 2022) which learns image-dependent caption prompts and prompt tuning (Lester et al., 2021; Liu et al., 2021a) which learns domain adaptation, our method employs task-level instructional prompts to approximate optimal text prompts for each image in a specific task, transforming text-to-image

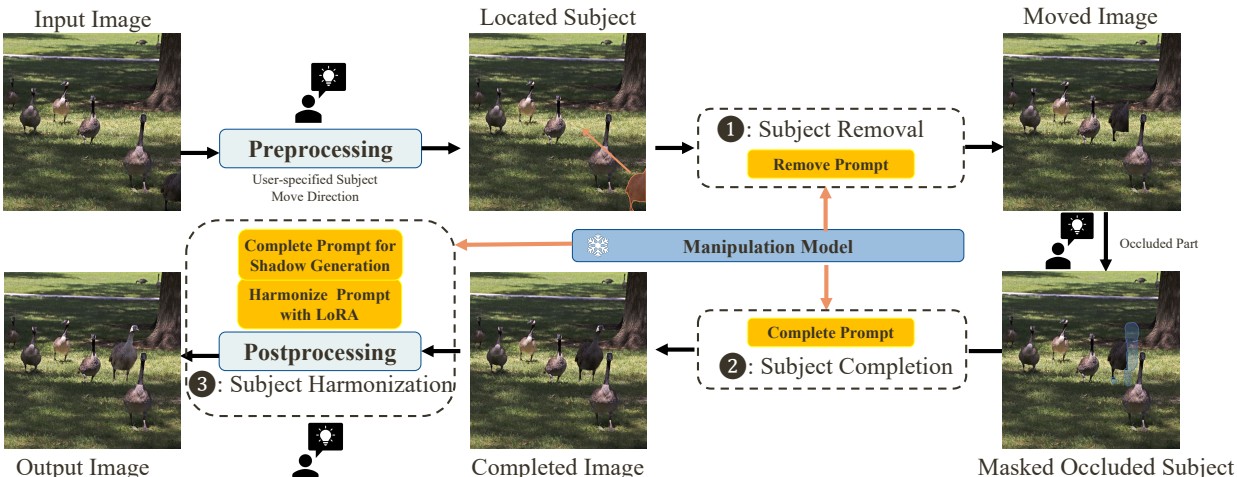

Figure 2: SEELE for SubRep includes i) pre-processing: identifying the subject via user-provided conditions, and preserving occlusion relationships between subjects; ii) manipulation: filling gaps left in the image and corrects obscured subjects with user-specified incomplete masks; iii) post-processing: addressing disparities between the repositioned subject and its new surroundings. SEELE addresses all generative sub-tasks in SubRep via a single diffusion model. In this example, only local harmonization is used in postprocessing. See shadow generation results in Figure 8.

diffusion model into task-to-image model. Our approach pioneers the systematic use of learned embeddings across generative sub-tasks within a single SD, effectively addressing the complex challenge of SubRep.

To formally address the SubRep task, we propose the SEgment-gEnerate-and-bLEnd (SEELE) framework. As in Figure 2, SEELE manages the subject repositioning with a pre-processing, manipulation, post-processing pipeline. i) In the pre-processing stage, SEELE segments the subject based on user-specified points, bounding boxes, or text prompts. With the provided moving direction, SEELE relocates the subject while considering occlusion relationships between subjects. ii) In the manipulation stage, SEELE uses a single diffusion model guided by learned task prompts to handle subject removal and completion. iii) In the post-processing stage, SEELE harmonizes the repositioned subject to blend with adjacent regions.

We've curated a dataset named ReS to test subject repositioning algorithms in real-world scenarios. We made efforts in covering various scenes and times to give a wide range of examples. Particularly, the real-world images for this task demand very exhaustively ground-truth annotation, including the mask of the repositioned subject and the moving direction. We annotate the mask using SAM (Kirillov et al., 2023) and manual refinement, and estimating the moving direction based on the center point of masks in the paired image. Additionally, we also provide amodal masks for subjects that are partly hidden. This results $100 \times 2$ paired real image, actually diverse enough to support the evaluation of our task, as illustrated in Figure 3(b). As far as we know, this is the first dataset designed specifically for subject repositioning. It's diverse and well-organized, making it a great benchmark for validating methods for this task.

**Contributions** Our contributions are as follows:

- We delineate the Subject Repositioning (SubRep) task as a specialized interactive image manipulation challenge, decomposed into several distinct sub-tasks, each of which presents unique challenges and necessitates specific capacities.
- We present the SEgment-gEnerate-and-bLEnd (SEELE) framework, which tackles various generative tasks using a single diffusion model. SEELE offers an application akin to Google's magic editor. Additionally, SEELE goes beyond the Magic Editor by offering advanced features like preserving occlusion and perspective, as well as local harmonization.
- We present task inversion, demonstrating that we can re-formulate the text-conditions to represent task instructions. This exploration opens up new possibilities for adapting diffusion models to specific tasks.
- We curate the ReS dataset, a real-world collection featuring repositioned subjects, serving as a benchmark for evaluating subject repositioning algorithms.

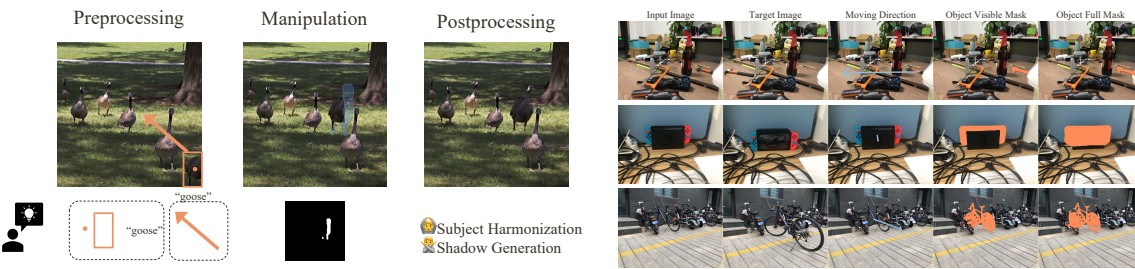

(a) User inputs in each stage.

(b) Examples of ReS dataset.

Figure 3: (a) User inputs in each stage of SubRep. (b) Examples of Res dataset. We provide paired images with subject full and visible mask annotations as well as moving direction information. The moving direction is marked as blue. The mask of visible part and completed subject specified by user are marked as orange.

## 2 Subject Repositioning

Subject repositioning (SubRep) relocates the user-specified subject within an image. The "subject" can be anything the user focuses on, such as a part of an object, an entire object, or multiple objects. Although it sounds straightforward, this task is quite complex. It requires coordination of multiple sub-tasks and interaction between user and learning models.

**User inputs** An illustration of the user inputs is shown in Figure 3(a). SubRep follows user intention to identify the subject, move it to the desired location, complete it, and address disparities. Particularly, the user identifies the interested subject via pointing, bounding box, or text prompts. Then, the user provides the desired repositioning location via dragging or direction. The user also needs to specify the occluded part of the subject for completion, and decide whether to apply post-processing to reduce visible differences.

**ReS dataset** To evaluate the effectiveness of subject repositioning algorithms, we curated a benchmark dataset called ReS. It includes $100 \times 2$ paired images: one image features a repositioned subject while the other elements remain constant. These images were collected from over 20 indoor and outdoor scenes, featuring subjects from over 50 categories. This diversity enables effective simulation of real-world applications, making our dataset suitable for evaluating our SEELE model.

We also contribute very detailed annotations to this dataset. Particularly, The masks for the repositioned subjects were initially generated using SAM and refined by multiple experts. Occluded masks were provided for subject completion. The direction of repositioning was estimated by measuring the distance between the center points of the masks in each image pair. For each paired image in the dataset, we can assess subject repositioning performance from one image to the other and in reverse, resulting in double testing examples. Figure 3(b) illustrates the ReS dataset. We release the ReS dataset at `https://yikai-wang.github.io/seele/` to encourage research in subject repositioning.

## 3 SEELE Framework for Subject Repositioning

**Task decomposition** To tackle this task, we introduce the SEgment-gEnerate-and-bLEnd (SEELE) framework, shown in Figure 2. Specifically, SEELE breaks down the task into three stages: pre-processing, manipulation, and post-processing. Pre-processing handles non-generative tasks, while manipulation and post-processing require generative capabilities. We use a unified diffusion model for all generative sub-tasks and pre-trained models for non-generative tasks in SEELE.

i) The *pre-processing* addresses how to precisely locate the specified subject with minimal user input, considering that the subject may be a single object, part of an object, or a group of objects identified by the user's intention; reposition the identified subject to the desired location; and also identify occlusion relationships to maintain geometric consistency. Additionally, adjusting the subject's size might be necessary to maintain the perspective relationship.

ii) The *manipulation* stage deals with the main tasks of creating new elements in subject repositioning to enhance the image. In particular, this stage includes the subject removal step, which fills the empty space on the left void of the repositioned subject. Additionally, the subject completion step involves reconstructing any obscured parts to ensure the subject is fully formed.

iii) The *post-processing* stage focuses on minimizing visual differences between the repositioned subject and its new surroundings. This involves fixing inconsistencies in both appearance and geometry, including blending unnatural boundaries, aligning illumination statistics, and, at times, creating realistic shadows.

**Pre-processing**  For point and bounding box inputs for identifying subject, we utilize SAM (Kirillov et al., 2023) for user interaction and employ SAM-HQ (Ke et al., 2023) to enhance the quality of segmenting subjects with intricate structures. To enable text inputs, we follow SeMani (Wang et al., 2023) to indirectly implement a text-guided SAM mode. Specifically, we first employ SAM to segment the entire image into distinct subjects. Then we identify the most similar one using the mask-adapted CLIP (Liang et al., 2022).

After identifying the subject, SEELE follows user intention to reposition the subject to the desired location, and masks the original area.

SEELE handles the potential occlusion between the moved subject and other elements in the image. If there are other subjects present at the desired location, SEELE employs the monocular depth estimation algorithm MiDaS (Ranftl et al., 2020) to discern occlusion relationships between subjects. SEELE will then appropriately mask the occluded portions of the subject if the user wants to preserve these occlusion relationships. MiDaS is also used to estimate the perspective relationships among subjects and resize the subject accordingly to maintain geometric consistency. For subjects with ambiguous boundaries, SEELE incorporates the ViTMatte matting algorithm (Yao et al., 2023) for better compositing with surrounding areas. An illustrated comparison of incorporated modules can be found in Figure 8.

**Manipulation**  In this stage, SEELE deals with the primary tasks of manipulating subjects, including subject removal and subject completion, as illustrated in Figure 2. Critically, such two steps can be effectively solved by a single generative model, as the masked region of both steps should be filled in to match the surrounding areas. However, these two sub-tasks require different information and types of masks. Particularly, for subject removal, a *non-semantic* inpainting is applied uniformly from the unmasked regions, using a typical object-shaped mask. This often falsely results in the creation of new, random subjects within the holes. On the other hand, subject completion involves *semantic-rich* inpainting and aims to incorporate the majority of the masked region as part of the subject. Critically, to adapt the same diffusion model to the different generation directions needed for the above sub-tasks, we propose the task inversion technique in SEELE. This technique guides the diffusion model according to specific task instructions. Thus, with the learned *remove-prompt* and *complete-prompt*, SEELE tackles these sub-tasks via a single generative model. An illustrated comparison between different task-prompts can be found in Figure 7(a).

**Post-processing**  In the final stage, SEELE blends the repositioned subject with its surroundings by tackling two challenges below. The illustrated comparison of post-processing can be found in Figure 8.

i) *Local harmonization* ensures natural appearance in boundary and lighting statistics. SEELE confines this process to the relocated subject to avoid affecting other image parts. It takes the image and a mask indicating the subject's repositioning as inputs. However, the stable diffusion model is initially trained to generate new contents within the masked region, conflicting with our goal of only ensuring consistency in the masked region and its surroundings. To address this, SEELE adapts the model by learning a *harmonize-prompt* with LoRA adapter (Hu et al., 2021) to guide masked regions. This can also be integrated into the same diffusion model used in the manipulation stage with our newly proposed design.

ii) *Shadow generation* aims to create realistic shadows for repositioned subjects, enhancing the realism. Generating high-fidelity shadows in high-resolution images of diverse subjects remains challenging. SEELE uses the diffusion model for shadow generation, addressing two scenarios: 1) If the subject already has shadows, we use *complete-prompt* for shadow completion. 2) For subjects without shadows, we follow user-intention to locate the desired shadow area. This task then transforms into a local harmonization process.

### 3.1 Task Inversion

Generative sub-tasks in subject repositioning follows the inputs and outputs of general inpainting task but with specific target:

1) **Subject removal** fills the void in original area without creating new subjects;

2) **Subject completion** completes the repositioned subject within masked region;

3) **Subject harmonization** blends subject without inducing new elements.

These requirements lead to different generation paths. Our goal is to adapt frozen text-to-image diffusion inpainting models for all of these sub-tasks.

**Task prompts** To address these challenges, we introduce *task inversion*, a method that trains prompts to guide the diffusion model for specific generation tasks while keeping its backbone fixed.

In standard text-to-image diffusion models, text prompts like "a cute cat" are processed through a text encoder, which generates token sequences to guide image generation. Task inversion eliminates the need for text inputs and the text encoder. Instead, we train learnable prompts, called *task prompts*, to serve as input sequences. These task prompts directly guide the model to perform specific tasks. Conceptually, they act as instructions such as "complete the subject". See Figure 4(a) for an illustration.

A key challenge is the domain gap: text-to-image diffusion models are not originally trained to respond to instruction-based prompts. However, our experiments demonstrate that learned task prompts significantly enhance performance. Compared to unconditional generation or simple semantic and instructional text prompts, task prompts deliver substantial improvements in standard inpainting and outpainting tasks (see Table 2) and sub-tasks like subject repositioning (see Table 1).

Our approach also reduces user effort by serving as an alternative to image-dependent text prompts for subject repositioning. Moreover, task inversion seamlessly integrates various generative sub-tasks for subject repositioning using stable diffusion. This eliminates the need for new generative models or extensive additional modules, emphasizing its plug-and-play simplicity.

**Inverse to learn task prompt** We aim to learn an optimal task prompt that conditions diffusion models for specific inpainting tasks. This task prompt is trained on input-output pairs, enabling it to translate task instructions from the training dataset into learned representations.

In text-to-image diffusion inpainting models, input conditions are typically text strings embedded into a sequence of vectors. For instance, in SD 2.0, a text encoder processes the text into a sequence of size $[L, D]$, which is then passed through the U-Net's cross-attention layers. Instead of using a text-based approach, our method directly learns a sequence of size $[L, D]$ to represent the task prompts.

Unlike user-driven prompts for specifying object location or direction, these task prompts are pre-learned during training. Each prompt is associated with specific input-output pairs tailored for the task, as explained in Sec. 3.2. During inference, task prompts are integrated into the pipeline shown in Figure 2. Users can then select between subject completion or harmonization via a simple button.

Formally, task inversion adheres to the original training objectives of diffusion models. Specifically, denote the training image as $\boldsymbol{x}$, the local mask as $\boldsymbol{m}$, the learnable task prompt as $\boldsymbol{z}$. Our objective is

$$\mathcal{L}(\boldsymbol{z}) \coloneqq \mathbb{E}_{\boldsymbol{\varepsilon} \sim \mathcal{N}(0,1), t \sim \mathcal{U}(0,1)} \left[ \| \boldsymbol{\varepsilon} - \boldsymbol{\varepsilon}_\theta([\boldsymbol{x}_t, \boldsymbol{m}, \boldsymbol{x} \odot (1 - \boldsymbol{m})], t, \boldsymbol{z} \|_{\mathrm{F}}^2 \right], \tag{1}$$

where $\boldsymbol{\varepsilon}$ is the random noise; $\boldsymbol{\varepsilon}_\theta$ is the diffusion model, $t$ is the normalized noise-level; $\boldsymbol{x}_t$ is the noised image, $\odot$ is element-wise multiplication; and $\| \cdot \|_{\mathrm{F}}$ is the Frobenius norm. When training with Eq. (1), the $\boldsymbol{\varepsilon}_\theta$ is frozen, making the embedding $\boldsymbol{z}$ the only learnable parameters.

Our task inversion is a distinctive approach, influenced by various existing works but with clear differences. The instruction prompt mentioned for our task inversion goes beyond the training data's scope, where the text describes the content of image, potentially affecting the desired generation results in practice. Recent advancements in textual inversion (Gal et al., 2022) emphasize the potential to comprehend user-specified concepts within the embedding space. In contrast, prompt tuning (Lester et al., 2021; Liu et al., 2021a)

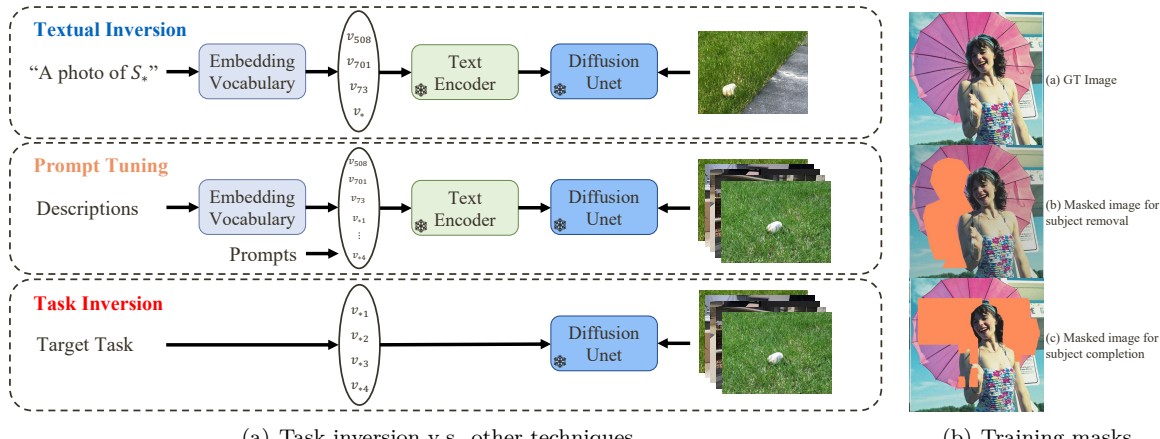

(a) Task inversion v.s. other techniques.      (b) Training masks.

Figure 4: (a) Comparison between task inversion and other techniques. Task inversion does not require text inputs, addresses different objectives, and serves different tasks, thus differing from other approaches. The embeddings $v_*$ and $v_{*i}$ are learnable and represented as $\boldsymbol{z}$ in Eq. (1). (b) We generate masks to represent particular tasks to train task inversion, addressing different tasks with a single diffusion model.

enhances adaptation to specific domains by introducing learnable tokens to the inputs. Unlike textual inversion, which trains a few tokens for visual understanding, our task inversion trains the whole latent to provide task instruction. Our task inversion differs prompt-tuning in that: prompt-tuning adds new tokens, while our approach replaces text condition inputs. We don't depend on text inputs to guide the diffusion model. See Figure 4(a) for the distinction.

### 3.2 Learning task inversion

Existing inpainting model is trained with randomly generated masks to generalize in diverse scenarios. In contrast, task inversion involves creating task-specific masks during training, allowing the model to learn specialized task prompts.

i) *Generating masks for subject removal*: In subject repositioning, the mask for the left void mirrors the subject's shape, but our goal isn't to generate the subject within the mask. To create training data for this scenario, for each image, we randomly choose a subject and its mask. Next, we move the mask, as shown by the girl's mask in the center of Figure 4(b). This results in an image where the masked region includes random portions unrelated to the mask's shape. This serves as the target for subject removal, with the mask indicating the original subject location and the ground-truth is background areas.

ii) *Generating masks for subject completion*: In this phase, SEELE addresses scenarios where the subject is partially obscured, with the goal of effectively completing the subject. To integrate this prior information into the task prompt, we generate training data as follows: for each image, we randomly select a subject and extract its mask. Then, we randomly choose a continuous portion of the mask as the input mask. Since user-specified masks are typically imprecise, we introduce random dilation to include adjacent regions within the mask. As illustrated by the umbrella mask on the right side of Figure 4(b), such a mask serves as an estimate for the mask used in subject completion.

iii) *Learning subject harmonization.* In SEELE, we achieve subject harmonization by altering the target of diffusion model. To this end, we take as input the inharmonious image and take as output the harmonious image. Additionally, we replace the unmasked region condition with original inharmonious image. Task prompt mainly influences the cross-attention layers. To adapt the self-attention in the diffusion model to preserve the content of masked region while harmonizing appearance, we introduce LoRA adapters (Hu et al., 2021). Our training objective is:

$$\mathcal{L} := \mathbb{E}_{\boldsymbol{\varepsilon} \sim \mathcal{N}(0,1), t \sim \mathcal{U}(0,1)} \left[ \| \boldsymbol{\varepsilon} + \boldsymbol{x} - \boldsymbol{x}^* - \boldsymbol{\varepsilon}_\theta([\boldsymbol{x}_t, \boldsymbol{m}, \boldsymbol{x}], t, \boldsymbol{z} \|_{\mathrm{F}}^2 \right], \tag{2}$$

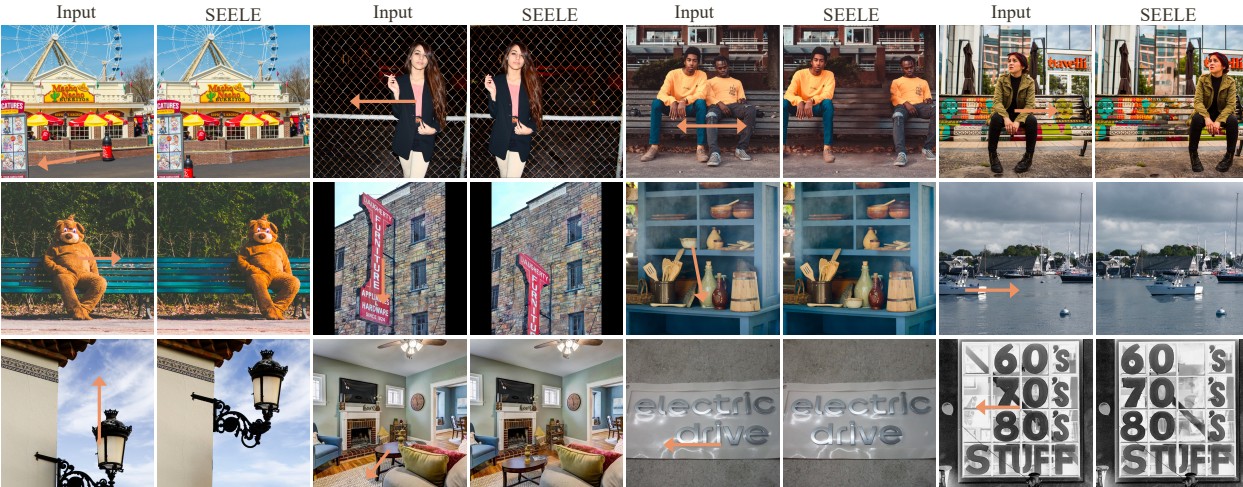

Figure 5: Subject repositioning on $1024^2$ images. SEELE works well on diverse scenarios, enabling flexible repositioning, and achieves high-fidelity repositioned images. Larger version in Figure 10.

where $\boldsymbol{x}^*$ represents the target harmonized image, and $\boldsymbol{x}$ is the input inharmonious image. This allows the diffusion model to gradually harmonize the image during denoising. While we modify the training objective, the generation process remains unchanged. This allows us to still utilize the pre-trained stable diffusion model with the learned harmonize-prompt and LoRA parameters, and seamlessly integrate with other modules.

## 4  Experimental Results and Analysis

**Examples of subject repositioning**   We present subject repositioning results on real-world $1024^2$ images using SEELE in Figure 5. SEELE works well on diverse scenarios, enabling flexible repositioning, and achieves high-fidelity repositioned images.

**Competitors and setup on ReS**   Google Photos' Magic Editor isn't publicly accessible, so we can't compare it with our method. We mainly compare with original Stable Diffusion inpainting model (SD) v2.0. We test SD with different prompts, including i) $SD_{no}$ performs unconditional generation; ii) $SD_{simple}$ uses "inpaint" and "complete the subject"; iii) $SD_{complex}$ uses "Incorporate visually cohesive and high-fidelity background and texture into the provided image through inpainting" and "Complete the subject by filling in the missing region with visually cohesive and high-fidelity background and texture" for subject removal and completion tasks, respectively. iv) $SD_{LoRA}$ uses the LoRA fine-tuning strategy to fine-tune the SD at the same training setup of SEELE. Furthermore, we can incorporate alternative inpainting algorithms in SEELE. Specifically, we incorporate LaMa (Suvorov et al., 2021), MAT (Li et al., 2022), MAE-FAR (Cao et al., 2022), and ZITS++ (Cao et al., 2023) into SEELE. We resize images to 512 pixels minimum for compatibility with standard inpainting algorithms. *Note that in this experiment, SEELE does not utilize any pre-processing or post-processing techniques. Standard inpainting algorithms cannot tackle subject repositioning without the incorporation of SEELE.*

**Qualitative comparison**   We present qualitative comparison results in Figure 6 where a larger version and more results are in the appendix. We add orange subject removal mask and blue subject completion mask in the input image. The SD column is SD guided by simple prompt as this variant performs best. Our qualitative analysis indicates that SEELE exhibits better subject removal capabilities without adding random parts and excels in subject completion. When the moved subject overlaps with the left void, SD fills the void by extending the subject. In contrast, SEELE avoids the influence of the subject, as in the top row of Figure 6. If the mask isn't precise, SEELE works better than other methods by reducing the impact of unclear edges and smoothing the area, as in the fourth row. SEELE excels in subject completion than typical inpainting algorithms, as in the second-to-last row. Note that *SEELE can be enhanced through the post-processing stage.*

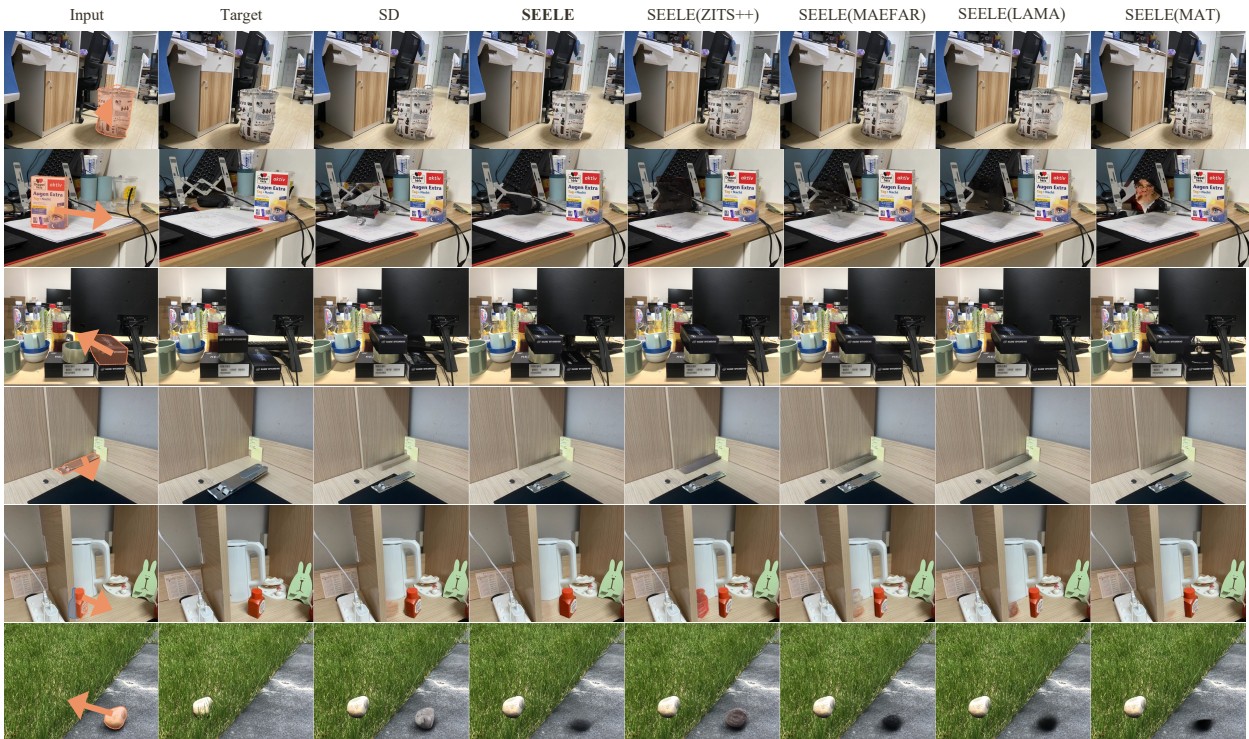

Figure 6: Qualitative comparison of subject repositioning on ReS. We add orange subject removal mask and blue subject completion mask in the input image. SEELE works better in the diverse real-world scenarios, even if the mask is not precise. Note that SEELE can be enhanced through the post-processing stage.

Table 1: Quantitative comparison and user-study on ReS. (∘): SD; (*): SEELE; Quality: the fidelity of the results; Consist.: the consistency with surrounding area. SEELE consistently works better than SD variants.

| Model | $\circ_{no}$ | $\circ_{simple}$ | $\circ_{complex}$ | $\circ_{LoRA}$ | SEELE | $*_{ZITS++}$ | $*_{MAE\text{-}FAR}$ | $*_{LaMa}$ | $*_{MAT}$ |
|---|---|---|---|---|---|---|---|---|---|
| LPIPS(↓) | 0.157 | 0.157 | 0.157 | 0.162 | **0.156** | 0.176 | 0.172 | 0.163 | 0.163 |
| Quality(↑) | 0.057 | 0.090 | 0.073 | 0.207 | **0.290** | 0.080 | 0.053 | 0.073 | 0.076 |
| Consist.(↑) | 0.054 | 0.057 | 0.050 | 0.036 | **0.329** | 0.089 | 0.114 | 0.168 | 0.104 |

**Quantitative comparison and user-study** We use Learned Perceptual Image Patch Similarity (LPIPS) as quantitative metric and conduct user-study to evaluate user preference from i) quality: the fidelity of the results; ii) visual-consistency (Consist.): the consistency with surrounding area. Our user study on all ReS dataset involves 100 anonymous surveys, reporting the ratio of top-1 preferred option. Results are in Table 1. Compared with other methods, SEELE demonstrates significant enhancements in the quality of manipulated images across all metrics. Particularly for the $SD_{LoRA}$, i) our construction of training mask requires object-level ground-truth segmentation in the training dataset, while public dataset do not have large scale annotated dataset (compared with the LAION dataset (Schuhmann et al., 2022) used by SD which contains 5B training data.) ii) when the training dataset is limited, the task inversion enjoys superior performance while fine-tuning technique leads to over-fitting and cause worse performance.

**Effectiveness of the proposed task-inversion** To further validate the proposed task-inversion, we conduct experiments on standard inpainting task on Places2 (Zhou et al., 2017) and outpainting task on Flickr-Scenery (Cheng et al., 2022), following the standard training and evaluation principles. Quantitative results is in Table 2, showcasing the superiority of the proposed task-inversion on both inpainting and outpainting tasks. We provide details and qualitative results in the appendix.

**Influence of different task prompts** We train different task prompts to guide different generation direction. Using wrong prompts for tasks can make the model give bad results. We tested this by comparing results from different learned task prompts. As in Figure 7(a), using a wrong prompt can change the outcome.

Table 2: Inpainting and outpainting comparison. Our task inversion achieves consistently better performance on standard inpainting and outpainting tasks. See qualitative comparison in the appendix. bkg: background, NA: no prompt.

(a) Inpainting on Places2 (Zhou et al., 2017).

| Methods | PSNR↑ | SSIM↑ | FID↓ | LPIPS↓ |
|---------|-------|-------|------|--------|
| Co-Mod | 21.09 | 0.84 | 30.04 | 0.17 |
| MAT | 20.68 | 0.84 | 32.44 | 0.16 |
| SD("NA") | 20.35 | 0.84 | 29.63 | 0.16 |
| SD("bkg") | 20.59 | 0.84 | 29.31 | 0.16 |
| SEELE | **21.98** | **0.87** | **24.40** | **0.13** |

(b) Outpainting on Flickr-Scenery (Cheng et al., 2022).

| Methods | SD("NA") | SD("bkg") | SEELE |
|---------|----------|-----------|-------|
| PSNR↑ | 14.48 | 14.60 | **16.00** |
| SSIM↑ | 0.69 | 0.70 | **0.73** |
| FID↓ | 53.52 | 46.58 | **29.06** |
| LPIPS↓ | 0.35 | 0.34 | **0.31** |

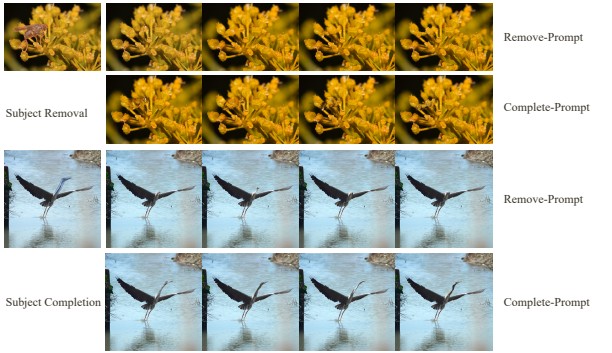

(a) Ablation of different task prompts.

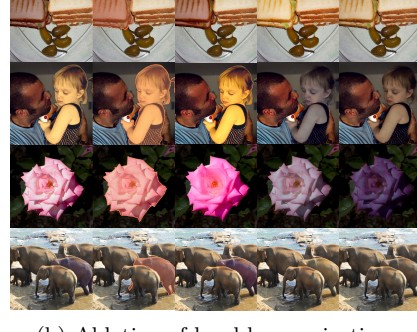

(b) Ablation of local harmonization.

Figure 7: (a) Opposite task prompts cause bad results. Zoom in to find the fly in 2rd-row. Different task prompt will lead to different generation direction. Use these prompt in the opposite way will cause bad results. (b) The local harmonization can be properly addressed with both the harmony-prompt along with the LoRA parameters.

For subject removal, remove-prompt can correctly generate with background flowers, while complete-prompt wrongly try to add a fly instead of flowers. For subject completion example of trying to add a bird's head, remove-prompt only added water, but the complete-prompt added the bird's head properly. This validate the different generation direction learned by our task prompt.

One might also want to use the LoRA technique to adapt the diffusion model for subject removal and completion. We run the experiments and compare this variant with only using the task inversion prompts. As shown in Tab. 3 (b), the LoRA-fine-tuned variant performs poorly. This shows that: (1) The subject removal and completion tasks are similar to generalized inpainting tasks, meaning that simply training task-specific prompts can effectively guide the diffusion model for these sub-tasks. (2) With the same training setup (using COCO), LoRA fine-tuning may reduce the U-Net's generalization ability. In contrast, our SEELE method keeps the U-Net frozen, maintaining its generalization ability.

**Ablation of Local Harmonization**   To tackle the local harmonization sub-task, we learn the harmony-prompt along with the LoRA parameters. To show the efficacy of each module, we conduct an qualitative ablation study in Figure 7(b). Naturally, if we disable the LoRA parameters, as we use the inharmonious image as unmasked image condition for the stable diffusion model, the model tends to copy the image without significant modification. If we only use LoRA parameter, it works like the unconditional diffusion model to perform local harmonization, but usually performs over- or under- harmonization. Such a manner works to some extent, but can be enhanced with the learned harmony-prompt.

Another choice for local harmonization is using specific local harmonization models. However, the benchmark dataset iHarmony4 (Cong et al., 2020) is usually used to train and test on a image size of $256 \times 256$, which is smaller than the standarad working resolution in SD 2.0 of size $512 \times 512$. Furthermore, the local

Table 3: Further analysis of SEELE..

(a) Local harmonization comparison on iHarmony4.

| Methods | PSNR↑ | MSE↓ |
|---|---|---|
| DucoNet (256, reported) | 39.17 | 18.47 |
| DucoNet (512, reproduced) | 31.55 | 197.38 |
| SEELE (512) | 31.88 | 78.74 |

(b) LoRA on subject removal and completion.

| Methods | PSNR↑ | SSIM↑ | LPIPS↓ |
|---|---|---|---|
| SD (no prompt) | 20.038 | 0.664 | 0.157 |
| SELE(LoRA) | 19.616 | 0.662 | 0.162 |
| SEELE | **20.100** | **0.666** | **0.156** |

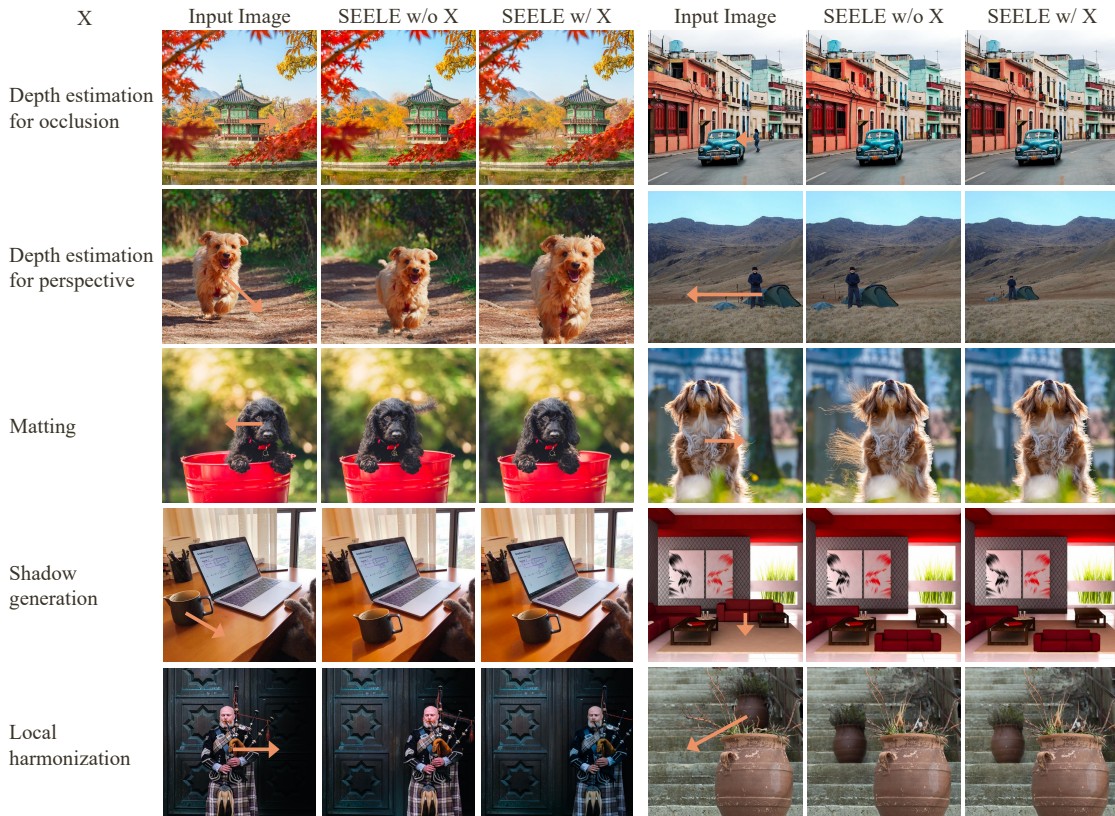

Figure 8: Ablation of using components X in SEELE. Applying specific component will lead to better consistency of generated images in corresponding perspective, and thus generating higher-fidelity images. See detailed analysis in the appendix.

harmonization models trained on smaller image sizes cannot generalize well on larger images. For instance, when we tested one of the SOTA models, DucoNet Tan et al. (2023), trained on iHarmony4, it didn't work as well as our model in our setup, as shown in Table 3(a). Hence we train image harmonization as part of our own framework. Since our framework is flexible, we can easily switch to a better harmonization model in the future if needed. This is also a benefit of our framework.

**SEELE w/ X** We assess the effectiveness of various components within SEELE during both pre-processing and post-processing phases. We conduct a qualitative comparison of SEELE's results with and without the utilization of these components, as in Figure 8, while a detailed analysis of is provided in the appendix.

**Failure analysis** As a sophisticated system, the success of SEELE relies on the success of each included module. Particularly, the core challenges of subject repositioning include appearance, geometry, and semantic inconsistency issues, as shown in Figure 9. i) SEELE addresses the appearance issue, which encompasses the absence of subjects and shadows, as well as unnatural shadows and boundaries. This is achieved through the innovative methods of subject completion, shadow generation, and local harmonization. ii) To tackle

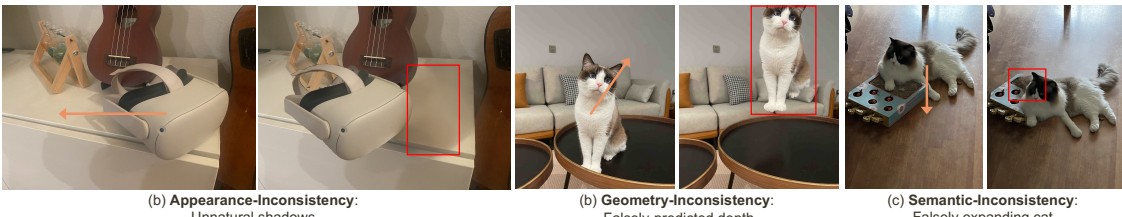

(b) **Appearance-Inconsistency**: Unnatural shadows.  (b) **Geometry-Inconsistency**: Falsely-predicted depth  (c) **Semantic-Inconsistency**: Falsely expanding cat.

Figure 9: Failure case visualization. The failure of modules in SEELE can lead to several inconsistencies.

the geometry issue, SEELE employs a depth estimation approach that maintains occlusion relationships and perspective accuracy. iii) For resolving semantic inconsistency, SEELE employs techniques for subject removal and completion. The failure of each specific module may lead to the corresponding inconsistency, and resulting in a less-fidelity image.

**Limitations** One significant limitation of SEELE is that when the system performs sub-optimally, manual user intervention becomes necessary to enhance the results. For instance, in cases where segmentation fails, users are required to manually correct the segment mask. Similarly, when the subject is occluded, users must provide a mask of potential regions to complete the subject. The former issue could potentially be mitigated through improvements in the segmentation model. However, the latter challenge necessitates the development of a novel model to address the problem of open-vocabulary amodal mask generation (Zhan et al., 2020). Currently, there lack available foundation models to support open-vocabulary amodal mask generation. These are potential avenues for future research.

## 5 Related Works

**Image and video manipulation** aims to manipulate images and videos in accordance with user-specified guidance. Among these guidance, natural language guidance, as presented in previous studies (Dong et al., 2017; Nam et al., 2018; Li et al., 2020a;b; Xia et al., 2021; Karras et al., 2019; El-Nouby et al., 2019; Zhang et al., 2021; Fu et al., 2020; Chen et al., 2018; Wang et al., 2018; Jiang et al., 2021), stands out as particularly appealing due to its adaptability and user-friendliness. Some research efforts have also explored the use of visual conditions, which can be conceptualized as image-to-image translation tasks. These conditions encompass sketch-based (Yu et al., 2019; Jo & Park, 2019; Chen et al., 2020; Kim et al., 2020; Chen et al., 2021; Richardson et al., 2021; Zeng et al., 2022), label-based (Park et al., 2019; Zhu et al., 2020; Richardson et al., 2021; Lee et al., 2020), line-based (Li et al., 2019), and layout-based (Liu et al., 2019) conditions. In contrast to image manipulation, video manipulation (Kim et al., 2019; Xu et al., 2019; Fu et al., 2022) introduces the additional challenge of ensuring temporal consistency across different frames, necessitating the development of novel temporal architectures (Bar-Tal et al., 2022) . Image manipulation primarily revolves around modifying static images, whereas video manipulation deals with dynamic scenes in which multiple subjects are in motion. In contrast, our paper focuses on subject repositioning, relocating one subject while the rest of the image remains unchanged.

**Textual inversion** (Gal et al., 2022) is designed to personalize text-to-image diffusion models according to user-specified concepts. It learns new concepts within the embedding space of text conditions while freezing other modules. Null-text inversion (Mokady et al., 2022) learns distinct embeddings at different noise levels to enhance capacity. Some fine-tuning (Ruiz et al., 2022) or adaptation (Zhang & Agrawala, 2023; Mou et al., 2023b) techniques inject visual conditions into text-to-image diffusion models. While these approaches concentrate on image patterns, SEELE focuses on the task instruction to guide diffusion models.

**Prompt tuning** (Lester et al., 2021; Liu et al., 2021b;a) entails training a model to learn specific tokens as additional inputs to transformer models, thereby enabling model adaptation to a specific domain without fine-tuning the model. This technique been widely used in vision-language models (Radford et al., 2021; Yao et al., 2021; Ge et al., 2022). This inspired us to adapt the text-to-image into task-to-image diffusion model by replacing the text conditions.

**Image composition** (Niu et al., 2021) is the process of combining a foreground and background to create a high-quality image. Due to differences in the characteristics of foreground and background elements, incon-

sistencies can arise in terms of appearance, geometry, or semantics. Appearance inconsistencies encompass unnatural boundaries and lighting disparities. Segmentation (Kirillov et al., 2023), matting (Xu et al., 2017), and blending (Zhang et al., 2020) algorithms can be employed to address boundary concerns, while image harmonization (Tsai et al., 2017) techniques can mitigate lighting discrepancies. Geometry inconsistencies include occlusion and disproportionate scaling, necessitating object completion (Zhan et al., 2020) and object placement (Tripathi et al., 2019) methods, respectively. Semantic inconsistencies pertain to unnatural interactions between subjects and backgrounds. While each aspect of image composition has its specific focus, the overarching goal is to produce a high-fidelity image. SEELE enhances harmonization capabilities within a single generative model.

**Drag-based manipulation** Pan et al. (2023) also performs dynamic manipulation on a static image. It works by selecting a point on the object, then dragging it to a new location to adjust the object's shape, direction, or pose to match the drag. To apply this to real images, the process usually involves reversing the image into an initial latent representation Pan et al. (2023) or noise Shi et al. (2024); Mou et al. (2023a); Luo et al. (2024); Mou et al. (2024); Liu et al. (2024). Features are extracted from this representation to recreate the image, and then manipulated to follow the drag direction. The manipulation is guided by methods like motion supervision Pan et al. (2023), explicit feature replacement Shi et al. (2024), or implicit gradient guidance Mou et al. (2023a); Luo et al. (2024); Liu et al. (2024). Fine-tuning is also used in some approaches to preserve the identity of the original image Shi et al. (2024). The key differences between drag-based manipulation and subject repositioning are: (1) Drag-based manipulation focuses on changing the shape of an object but does not handle subject completion, which is essential for subject repositioning. (2) Inversion-based methods struggle to preserve unchanged areas and the repositioned subject, while our approach regenerates only the necessary regions.

## 6 Conclusion

In this paper, we introduce an innovative task known as subject repositioning, which involves manipulating an input image to reposition one of its subjects to a desired location while preserving the image's fidelity. To tackle subject repositioning, we present SEELE, a framework that leverages a single diffusion model to address the generative sub-tasks through our proposed task inversion technique. This includes tasks such as subject removal, subject completion, and subject harmonization. To evaluate the effectiveness of subject repositioning, we have curated a real-world dataset called ReS. Our experiments on ReS demonstrate the proficiency of SEELE.

**Broader Impact Statement**

Our proposed SEELE system aims to address the issue of subject repositioning within single images and will be responsive to user intentions. However, there is a risk that it could be misused to create prank images with malicious intent towards individuals, entities, or objects. To mitigate this, we add watermarks to images generated by our SEELE system to indicate their artificial nature.

**Acknowledgments**

This work was supported in part by NSFC under Grant (No. 62076067). The computations in this research were performed using the CFFF platform of Fudan University.

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

# A Appendix

## A.1 Additional Examples

In this section, we first present subject repositioning results on images of size $1024 \times 1024$ using SEELE (Fig. 5 in our paper) in Figure 10. Furthermore, we present additional examples of subject repositioning using SEELE and its competitors, as showcased in the proposed ReS dataset, within Figure 11.

## A.2 Experimental Setting

SEELE is built upon the text-guided inpainting model fine-tuned from SD 2.0, employing the task inversion technique to learn each task prompt with 50 learnable tokens, initialized with text descriptions from the task instructions. For each task, we utilize the AdamW optimizer (Loshchilov & Hutter, 2017) with a learning rate of $8.0e - 5$, weight decay of 0.01, and a batch size of 32. Training is conducted on two A6000 GPUs over 9,000 steps, selecting the best checkpoints based on the held-out validation set.

When addressing subject moving and completion, we employ the MSCOCO dataset (Lin et al., 2014), which provides object masks. For image harmonization, the iHarmony4 dataset (Cong et al., 2020) is utilized, offering unharmonized-harmonized image pairs along with subject-to-harmonize masks. MSCOCO comprises 80k training images and 40k testing images, while iHarmony4 includes 65k training images and 7k testing

images. This diversity ensures robustness in training task prompts, guarding against overfitting on specific images.

**Cost analysis**  The core component of SEELE is the pre-trained stable diffusion inpainting model, boasting 865.93 million parameters within its UNet backbone. To tailor this stable diffusion model for subject repositioning, we incorporate three distinct task prompts, each sized at $50 \times 1024$ and has 0.5 million trainable parameters. For the local harmonization task, we introduce the LoRA adapter, which encompasses 5.12 million trainable parameters. It's worth noting that these newly added parameters are lightweight and introduce no additional inference latency when compared to the stable diffusion backbone.

### A.3  Analysis of X in SEELE

Here we provide the analysis of each component used in SEELE.

i) *Depth estimation for occlusion* becomes crucial when users wish to move a subject from the foreground to the background. It helps estimate and correct the occluded parts, ensuring that the repositioned subject blends seamlessly into the scene. As illustrated in the first row of Figure 8, this depth estimation plays a pivotal role in repositioning objects like the tower behind leaves or people behind a car. Neglecting the occlusion relationship can result in unnatural-looking repositioned subjects and a significant loss of image fidelity.

ii) *Depth estimation for perspective* comes into play when users want to resize the subject proportionally during repositioning. If this aspect is overlooked, the subject's size remains fixed, which may contradict user expectations.

iii) *Matting* primarily addresses issues arising from imprecise masks provided by SAM, particularly when dealing with subjects with ambiguous boundaries. Precise masking is crucial because inaccuracies can lead to information leaking in the final output. For example, in Figure 8, imprecise masking might encourage the gaps to generate unnatural dog fur.

iv) *Shadow generation* is handled by reusing the generative model within SEELE. In cases where a subject includes shadows, such as the left part in Figure 8, we approach it as a subject completion task. The shadow itself becomes the subject, and we employ a learned complete-prompt to guide the diffusion model. Conversely, when a subject lacks shadows, we can transform it into a local harmonization task by utilizing SEELE's harmonization model to generate shadows.

v) *Local harmonization* addresses the challenge of appearance inconsistency. When the illumination statistics change after subject repositioning, it's essential to adjust the subject's appearance while preserving its texture. As depicted in Figure 8, SEELE excels at this local harmonization task, ensuring seamless integration into the new environment.

### A.4  Standard Image Inpainting and Outpainting

**Image inpainting**  The proposed task-inversion approach not only specializes the inpainting model for specific tasks but also enhances its standard inpainting capabilities. We substantiate this claim through experiments conducted on the Places2 dataset (Zhou et al., 2017), where we train SEELE using standard inpainting prompts and compare its performance with other inpainting algorithms. The results are presented in Tab. 2(a) in our paper. Additionally, we provide visual representations of the results in Figure 12, demonstrating SEELE's advantage in reducing hallucinatory artifacts.

**Image outpainting**  Another commonly used manipulation task involves extending the image beyond its original content. This approach shares a similar concept with subject completion, but it takes a more holistic perspective by enhancing the entire image. We have also conducted experiments on the outpainting task and demonstrated the effectiveness of task inversion. Our experiments were carried out using the Flickr-Scenery dataset (Cheng et al., 2022), and the results are compared with stable diffusion in Tab. 2(b) in our paper. The results indicate the superiority of task inversion employed in SEELE. Furthermore, we provide visual examples for qualitative assessment in Figure 13.

### A.5 Necessity of Using Different Datasets to Train SEELE

Our training of the SEELE model utilized only two datasets: COCO, which provides ground-truth object segmentation masks, and iHarmony4, which offers paired images for local harmonization tasks. These datasets, chosen for their public availability, aptly fulfill the varying requirements of different generative sub-tasks. Our training approach, which encompasses both subject movement and completion, employs a unified task inversion technique. Given that local harmonization focuses on not introducing new details in masked areas, we have modified the diffusion model to integrate the characteristics of the masked region, ensuring it aligns with the task's specific needs.

### A.6 Integrating LoRA

When the LoRA adapter is trained, we load them along with the frozen stable diffusion model. As LoRA is implemented as additive layers with the original layers. For example, suppose for a particular layer $f$ with input $x_i$ and output $x_{i+1}$. The original stable diffusion performs $x_{i+1} = f(x_i)$, while LoRA is trained to perform $x_{i+1} = f(x_i) + \text{LoRA}(x_i)$ and only learn $\text{LoRA}(\cdot)$ while freezing $f(\cdot)$. Then we could introduce a scale hyper-parameter for a trained model $x_{i+1} = f(x_i) + c\text{LoRA}(x_i)$ When SEELE performs the sub-tasks in manipulation process, we set the lora scale as $c = 0$ to preserve the original outputs of stable diffusion. While in the local harmonization process, we set the lora scale as $c = 1$ to perform local harmonization. In this regard, we could use the same stable diffusion backbone and perform different sub-tasks using different sub-task prompts (and LoRA parameters).

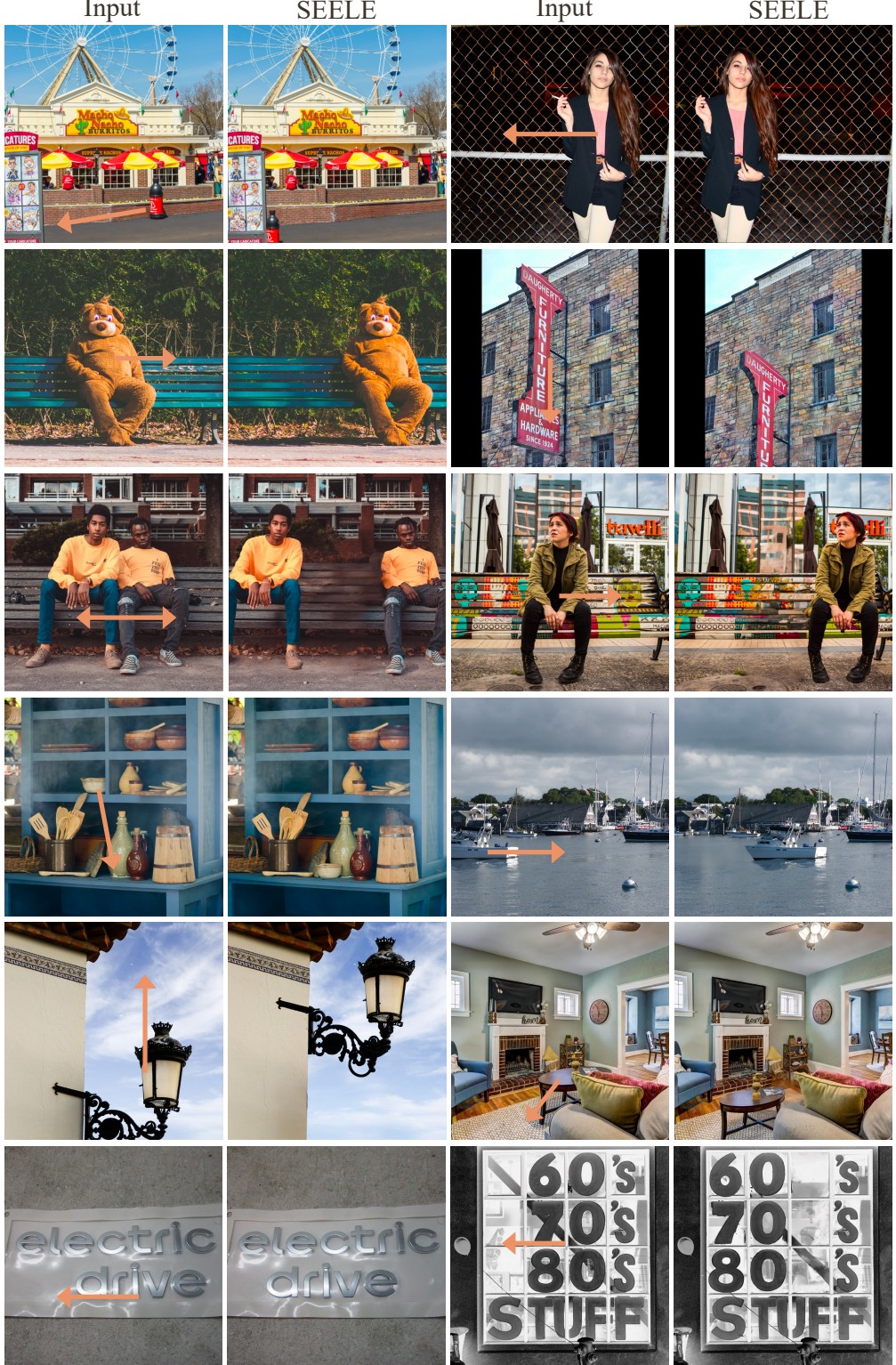

Figure 10: SEELE on images of size $1024 \times 1024$.

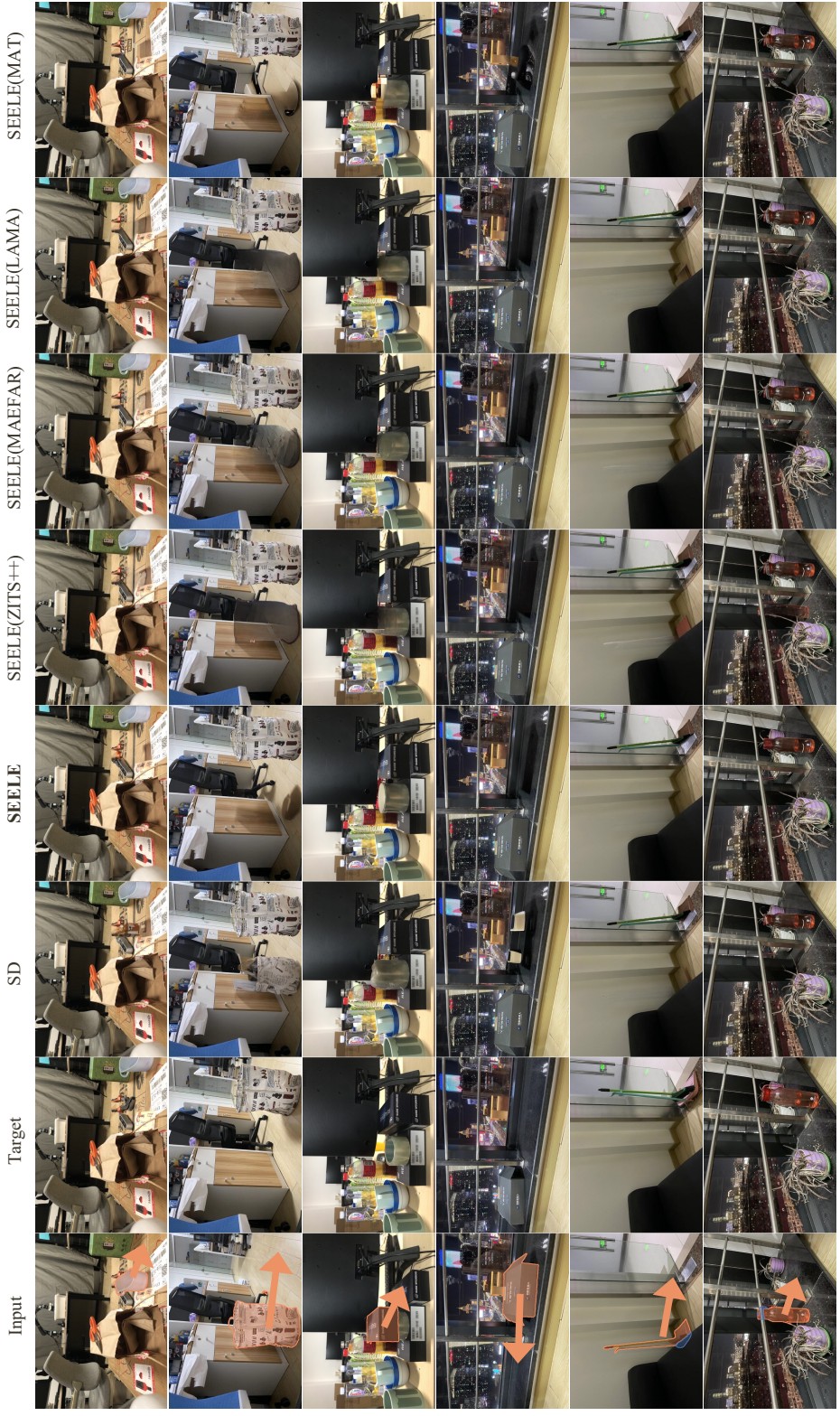

Figure 11: More qualitative comparison for subject repositioning in ReS.

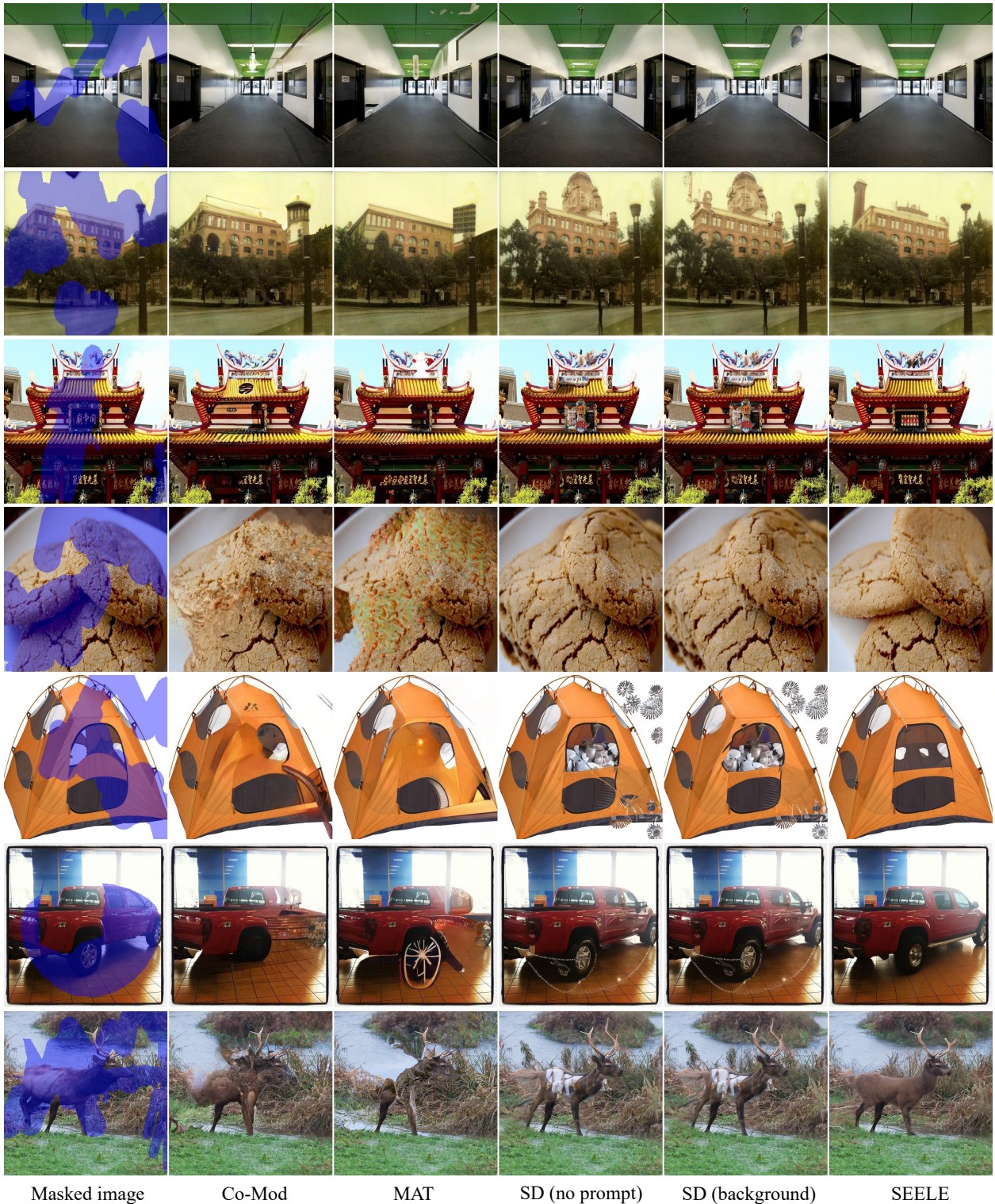

Figure 12: Qualitative comparison for inpainting.

SD

SEELE

SD

SEELE

SD

SEELE

SD

SEELE

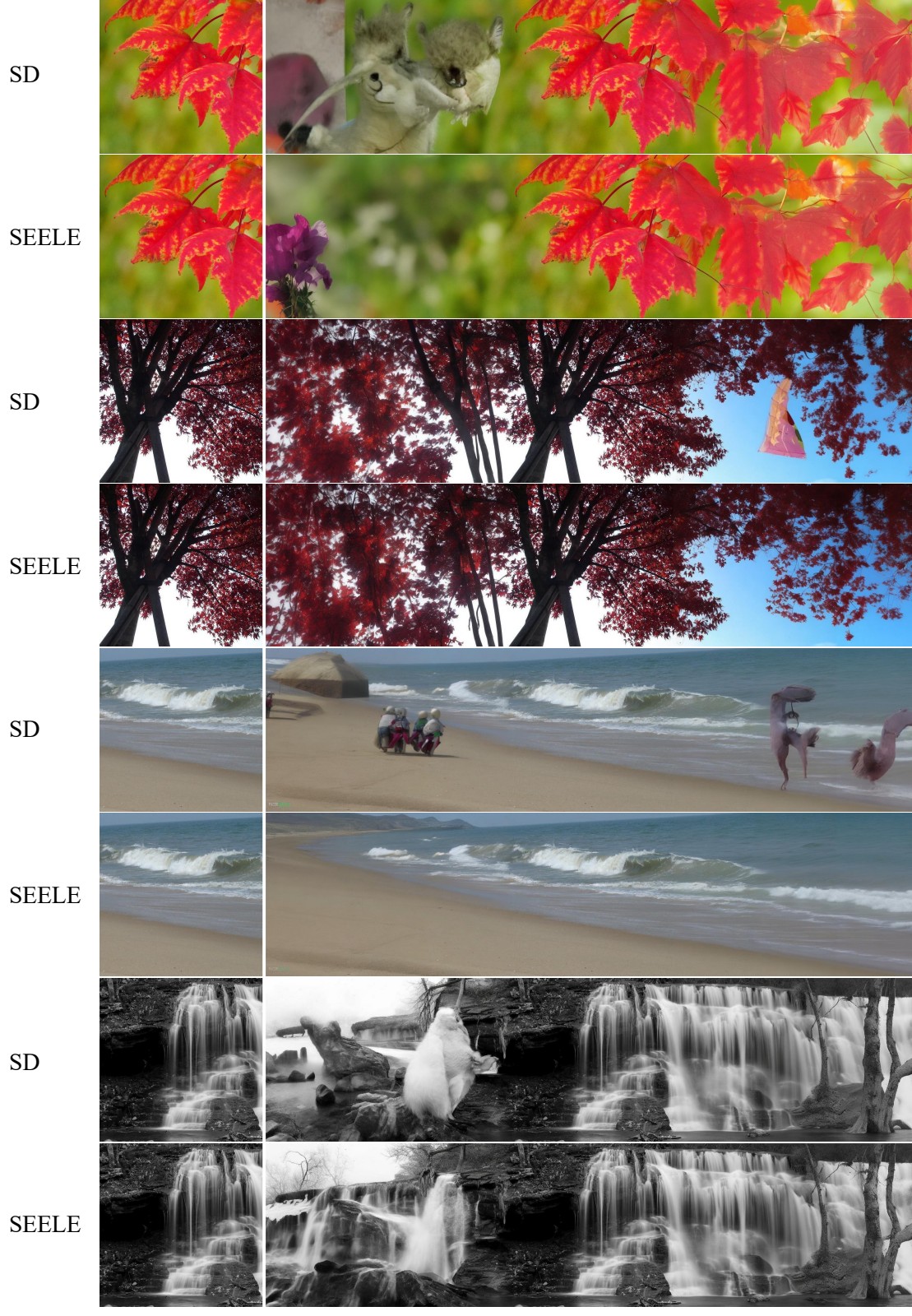

Figure 13: Qualitative comparison for outpainting.

