# OpenReview forum: "Repositioning the Subject within Image"
_TMLR — Accepted by TMLR_

### Review · Reviewer_4XUX · 2024-08-03

**Summary Of Contributions:**

This work proposes a subject repositioning system that integrates several components with preprocess and post-process. The subject repositioning task is decomposed into several distinct sub-tasks and each is handled with a proposed StableDiffusion based model. In particular, the object removal and completion are achieved via a ``task inversion’’ technique, and the local harmonization is achieved via lora finetuning technique. An evaluation benchmark, consisting of 100 × 2 paired real images, is also introduced. Experiment results show that the proposed method can produce good results on the subject repositioning task.

**Audience:**

Yes

**Broader Impact Concerns:**

A broader impact section is involved.

**Claims And Evidence:**

Yes

**Requested Changes:**

The lora based shadow generation and local harmonization implementation should be explained more clearly for re-implementation. A comparison of the local harmonization with prior works should be necessary.

I wonder if the author will the code and models.

**Strengths And Weaknesses:**

Pros:

- The system pipeline is carefully designed. Although it contains several modules, the pipeline design is reasonable and fits the practical application scenario.

- The task inversion seems similar to prior techniques, like textural inversion and prompt tuning, but without a text encoder and with a different target. The application of task inversion is insightful because different tasks can be solved while the SD backbone stays the same.

- The experiments are comprehensive, including qualitative and quantitative comparisons, as well as user studies.
The results look good compared to other baselines. The ablation studies of different task prompts, different inpaint modules, and w/wo local harmonization are also reported to clearly demonstrate the effectiveness of each component.

- The paper is easy to follow. The evaluation set can benefit the following works.


Cons:

- I believe drag-based image manipulation methods are quite related to this work, it would be necessary to discuss drag-based related works such as DragGAN,  DragDiffusion, DragonDiffusion, etc.

- The move direction and shadow generation are not illustrated in Figure 2, which is confusing.

- If I understand correctly, the incomplete mask used to complete the object is user-specified instead of generated automatically.  If so, this is the bottleneck of full automation.

- The lora based shadow generation and local harmonization are not clear. It is claimed to not add extra details to the local object. However, there is randomness during the diffusion process. The exact implementation should be included in the main paper. Furthermore, there should be some comparisons between the proposed lora based local harmonization and prior image harmonization works. If prior works perform better, why not just integrate prior harmonization works?

- If the  local harmonization can perform well via lora technique, I doubt how the object remove and completion perform via lora compared to the “task inversion”

- The background becomes blurry compared to Google Magic Editor (see Figure 1).

---

> ### Author Response · Authors · 2024-10-02
>
> **1: Discussing drag-based image manipulation methods.**
>
> Thank you for your suggestion. We will discuss them in our revisions. Specifically, we would like to highlight that many drag-based methods, like Dragondiffusion, work by inverting real images, but this makes it hard to precisely keep the unchanged areas and the repositioned subject. Additionally, these methods can't handle subject completion, which is a key part of our approach.
>
> **2: Suggestions about Figure 2.**
>
> Thank you for your suggestion. In our revisions, we will include the movement direction. We’ll also clarify that Figure 2 doesn't show shadow generation and refer to Figure 8 for those results.
>
> **3: The user-specified completion mask.**
>
> Thank you, and yes, the incomplete mask is provided by the user. We actually tried using several amodal mask completion models to automate this process, but they didn't perform well in our open-vocabulary setups. So, we decided not to include them in our framework.
>
> **4: About the shadow generation and local harmonization details.**
>
> Thanks for your suggestion. We presented the implementation details in Sec. 3.2, and would like to further refine it in our revision.
>
> We didn't use previous image harmonization methods because most of them were trained on smaller images (256x256) from the iHarmony4 dataset, while our backbone model (SD 1.5) uses a larger standard resolution (512x512). These methods don't perform well on larger images. For instance, when we tested one of the SOTA models, DucoNet [1], trained on iHarmony4, it didn't work as well as our model in our setup. That's why we decided to train image harmonization as part of our own framework.
> Since our framework is flexible, we can easily switch to a better harmonization model in the future if needed. This is also a benefit of our framework.
> | **Model**      | **PSNR(↑)**  | ***MSE(↓)**  |
> |----------------|-----------|----------|
> | DucoNet  (256, reported)  |   39.17   |18.47|
> | DucoNet  (512, reproduced)       | 31.55 |197.38|
> | SEELE            |31.88  |78.74|
>
>
> [1] Tan L, Li J, Niu L, et al. Deep image harmonization in dual color spaces[C]//Proceedings of the 31st ACM International Conference on Multimedia. 2023: 2159-2167.
>
> **5: Comparison with Lora based remove and completion.**
>
> Thank you for the suggestion. Based on it, we trained models for subject removal and completion using LoRA and compared them with task inversion prompts. The results are as follows:
> | **Model**      | **PSNR(↑)**  | **SSIM(↑)**  | **LPIPS(↓)**  |
> |----------------|-----------|-----------|-----------|
> | SD (no prompt)    | 20.038       |0.664|0.157|
> | SELE(Lora)           | 19.616       |0.662|0.162|
> | SEELE            | 20.100      |0.666|0.156|
> The LoRA-fine-tuned variant performs poorly. This shows that:
>
> The subject removal and completion tasks are similar to generalized inpainting tasks, meaning that simply training task-specific prompts can effectively guide the diffusion model for these sub-tasks.
>
> With the same training setup (using COCO), LoRA fine-tuning may reduce the U-Net's generalization ability. In contrast, our SEELE method keeps the U-Net frozen, maintaining its generalization ability.
>
> **6: Blurry generation results.**
>
> Thanks. Our framework uses the frozen SD1.5 inpainting model, meaning its ability to generate images depends entirely on SD1.5. We believe that using a stronger generation model would lead to better results. Please note that this paper doesn't focus on improving the generation ability of diffusion models. Instead, it adapts the frozen diffusion model for specific inpainting tasks without text input.
>
> **7. Release code and models:**
>
> We plan to release our code and models after acceptance.

---

### Review · Reviewer_knYi · 2024-08-13

**Summary Of Contributions:**

This paper focuses on the task of relocating objects in an image. It proposes the novel SEELE framework to solve this task with a single diffusion, and construct a ReS dataset for evaluation. The experiments show that the model can deal with various challenging and complicated tasks and produce high-quality images.

**Audience:**

Yes

**Broader Impact Concerns:**

The authors have discussed the broader impact in their paper.

**Claims And Evidence:**

Yes

**Requested Changes:**

Please answer the clarification questions mentioned in "Weaknesses," and try to improve the presentation.

**Strengths And Weaknesses:**

### Strength
- The idea of the method is novel and interesting.
- The quality of the images shown in the experiments is quite good.
- Abundant experiments and ablation studies show that the proposed method is powerful and can deal with various complicated and challenging situations.
- They construct the ReS dataset for evaluation.
### Weaknesses
- Why is the task called "subject" repositioning (removal, completion, harmonization) instead of "object" repositioning (removal, completion, harmonization)? What does "subject" mean here? It seems that the task is just moving objects in the image, and Fig.3(b) is also using "object" visible/full mask in wording.
- The "Task inversion" in Sec. 3.1 and Sec.3.2 is very confusing.
  - The only description of what task inversion is just as below, while the rest of the contents seems only to discuss the difference between task inversion and other tricks, *without* going into any details of the task inversion *itself*
    > _To address this challenge, we introduce task inversion, training prompts to guide the diffusion model while keeping the backbone fixed. Instead of traditional text prompts, we utilize the adaptable representations acting as instruction prompts, such as “complete the subject”._
   - Fig.4(a) is also very confusing. The way to obtain $v_{*1}, v_{*2}, ...$ (embeddings?) is never mentioned in the paper.
  - I wonder: (1) why is task inversion called "inversion," and what is it trying to inverse? (2) What is the form of "task prompt" as a prompt embedding that can be used by the diffusion model?  (3) How can the task prompt be calculated with user input of object location and direction?
- (Minor) Presentations
    - The rounded rectangle bounding boxes with dashed boundaries in Fig.1, Fig.2, Fig.3, Fig.4(a) do not have any semantic meaning, but are occluded with the texts.
    - In Tab.2, the name of the models should be in `\text` if indicating in LateX environment, e.g., $MAE-FAR$ should be $\text{MAE-FAR}$, the former means $M\times A\times E - F\times A\times R$.

---

> ### Author Response · Authors · 2024-10-02
>
> **1: Why use “subject” instead of “object”?**
>
> Thank you for your suggestion. We want to clarify that the subject is decided by the user, which can include part of an object, a whole object, or multiple objects. For example, in the first row of Figure 1, we move the child, the bench, and the balloons at the same time. Therefore, we prefer using the word “subject” to describe the areas of interest, rather than the narrower term “object.”
>
> **2-1: Definition of the task inversion.**
>
> Thank you for your question. In our original paper, the sentence you referred to provides an informal definition of our concept called task inversion. The main idea is to replace text prompts with learnable task prompts.
>
> In the next paragraph of the quoted sentence, we provide a mathematical formulation to explain the process of learning these task prompts. We plan to revise our manuscript to further clarify the definition of task inversion. Specifically, we will add the following explanation:
>
> "Task inversion involves replacing text prompts in text-to-image diffusion models. Normally, these models use text prompts like 'a cute cat' as input and then process them through a text encoder to create a sequence of tokens. These tokens guide the image generation process. Our task inversion removes the text encoder and the need for text-based input. Instead, we directly train learnable prompts, called task prompts, to serve as the input sequence. These task prompts guide the model to perform specific tasks."
>
> **2-2: Embeddings in Fig. 4(a).**
>
> Thanks for your question. The sequence $\[v_{*1},\ldots,v_{*4}\]$ is the learnable $z$ in Eq. (1). We will clarify this in our revision.
>
> **2-3: About the “inversion” and details of the task prompt.**
>
> We call our method "task inversion," inspired by "textual inversion." Textual inversion aims to learn an embedding that represents a concept in the text embedding space, based on one or a few images that illustrate the concept. Similarly, task inversion aims to learn a task prompt that acts as a condition for diffusion models. This task prompt is learned by training on input-output pairs to perform specific inpainting tasks, as described in Eq. (1). The reason we call it "task inversion" is that we're reversing the task instructions from the training dataset into learned task prompts.
>
> In the original SD 1.5 inpainting model, text strings are used as input conditions, which are embedded into a sequence of embeddings. A text encoder then processes these embeddings into a sequence of size [77, 768].
> This sequence is fed into the cross-attention layers of the U-Net in SD 1.5. In our approach, we directly learn a sequence of [77, 768] using Eq. (1) to generate the task prompts.
>
> It's important to note that the task prompts are not generated based on user input for object location and direction. Instead, during training, we create specific input-output pairs for each task prompt, as explained in Sec. 3.2. During inference, task prompts are used according to the pipeline in Figure 2, where the user can choose whether to apply subject completion or harmonization via a button.
>
> **3: Presentations.**
>
> Thanks for your suggestions. We will revise them accordingly in our revision.

---

### Review · Reviewer_97Sw · 2024-09-18

**Summary Of Contributions:**

This paper aims at the subject repositioning task in image editing. They introduce SEgment-gEnerate-and-bLEnd (SEELE) to fill the void left and blend the object with the surroundings. Specifically, SEELE designs pre-processing and post-processing techniques to further enhance the final quality via a single diffusion model.

**Audience:**

Yes

**Broader Impact Concerns:**

The proposed SEELE relies on a pre-trained diffusion model, which may propagate the potential error to the final results.

**Claims And Evidence:**

Yes

**Requested Changes:**

+ A discussion of the novelty of SEELE.
+ A baseline relied on SD-Inpaint-style models.

**Strengths And Weaknesses:**

**Strengths**
+ The SEELE pipeline is reasonable, which leverages SAM to segment the assigned object and performs inpainting via the diffusions model.
+ From Fig. 6, they show the potential scalability of SEELE, which can be model-agnostic and collaborate with external models such as LAMA and MAT.
+ They provide comprehensive qualitative comparisons in different aspects through Fig. 5-13.

**Weakness**
+ The novelty may be the main issue of this paper. The proposed SEELE seems to be a combination of existing components (SAM+inpainting) instead of fundamental research. Though it can bring satisfactory results, I am not sure if this can achieve TMLR's bar.
+ One important baseline is missed; after utilizing SAM to recognize the target object, we can use SD-Inpaint (e.g., https://huggingface.co/diffusers/stable-diffusion-xl-1.0-inpainting-0.1) to do both removing and blending and accomplish object repositioning.

---

> ### Author Response · Authors · 2024-10-02
>
> **1: Novelty.**
>
> Thank you for your concern. We would like to clarify that, according to the acceptance criteria of TMLR, "novelty of the studied method is not a necessary criteria for acceptance."
>
> **2: Compare with SD inpaint.**
>
> We would like to clarify that we have already compared our results with SD-inpaint, which aligns exactly with your description, in all of our experiments. You can find the details in the "Competitors and Setup on ReS" section.

---

### Decision · Action_Editor_CExa · 2024-10-27

**Recommendation:** Accept with minor revision

**Comment:**

This paper focuses on the subject repositioning task in image editing.The authors introduce the SEELE framework to fill the void left and blend the object with the surroundings, where a single diffusion model is used for different sub-tasks via task inversion. After author rebuttal, it received two Accept and one Leaning Accept recommendations. Generally, all the reviewers are happy about the paper, commenting that the system pipeline is carefully designed, the experiments are comprehensive, and the reported results are satisfactory.

There are some concerns that were well addressed during rebuttal. In the final version, the authors are expected to include (1) a more clear discussion of task inversion, (2) discussion on drag-based image manipulation methods in related work, (3) comparison and a better discussion of LoRA based methods for local harmonization and the use of task inversion for object removal and completion. Overall, this is a good paper, and I recommend Accept with minor revision to reflect the above changes requested by reviewers. One reviewer also mentioned that open-sourcing the code and model would be important for this paper, and the authors are highly encouraged to do so.

**Audience:**

Yes, researchers who are working in the field of image editing and generation could be interested in this paper.

**Claims And Evidence:**

Yes, it's well supported.